# The Key Role of Psychosocial Competencies in Evidence-Based Youth Mental Health Promotion: Academic Support in Consolidating a National Strategy in France

**DOI:** 10.3390/ijerph192416641

**Published:** 2022-12-11

**Authors:** Béatrice Lamboy, François Beck, Damien Tessier, Marie-Odile Williamson, Nadine Fréry, Roxane Turgon, Jean-Michel Tassie, Julie Barrois, Zinna Bessa, Rebecca Shankland

**Affiliations:** 1Santé Publique France, 94410 Saint-Maurice, France; 2Laboratoire SENS, Université Grenoble Alpes, 38000 Grenoble, France; 3Instance Regionale d’Education et de Promotion de la Santé, Pays de la Loire, 44000 Nantes, France; 4Laboratoire Interuniversitaire de Psychologie (LIP/PC2S), Université Grenoble Alpes-Université Savoie-Mont-Blanc, 38000 Grenoble, France; 5Direction Générale de la Santé, Ministère de la Santé et de la Prévention, 75000 Paris, France; 6Laboratoire DIPHE, Université Lumière Lyon 2, 69000 Lyon, France; 7Institut Universitaire de France, 75000 Paris, France

**Keywords:** psychosocial competencies, mental health promotion, prevention of mental disorders, well-being, evidence-based policies

## Abstract

Psychosocial competencies, also known as psychosocial skills or life skills, are essential for the prevention and promotion of mental health. Since the beginning of this century, psychosocial competencies have been defined as the ability to develop positive mental health. Most individual or social mental health protection programs are related to psychosocial competencies. A majority of evidence-based programs that develop mental health explicitly aim at developing psychosocial competencies, either exclusively or with complementary approaches. Many of these programs have demonstrated their effectiveness, with lasting effects on reduced anxiety and depression symptoms, violent and risky behaviors, and improved well-being and academic success. Based on international meta-analyses and on 20 years of French national and local experiences, a national strategy to develop psychosocial competencies was launched in France in 2021 for all children from 3 to 25 years old. Two reports on evidence-based psychosocial competence development were published in 2022 by the national agency for public health—Santé publique France (Public Health France)—to support this deployment strategy and develop a common evidence-based culture in health and education. This article presents the French national strategy as an example of a means of increasing evidence-based mental health promotion while discussing the importance of cultural adaptation of such programs.

## 1. Introduction

In the course of the Assises de la Santé Mentale et de la Psychiatrie (Mental Health and Psychiatry Convention) in France, in September 2021, organized while mental health was impacted by the COVID pandemic, it was announced that a multisector strategy would be developed for the deployment of psychosocial competencies (PSCs) of children and adolescents. Indeed, the COVID-19 health crisis was a reminder that mental health is a dimension of global health in its own right, inseparable from its physical dimension, thus echoing the WHO definition of health as defined in its 1948 constitution [1] and The Lancet, which in 2007 announced: “No health without mental health” [2].

The field of mental health therefore occupies a strategic place in public health today [3]. The field of prevention and promotion of mental health (PPMH) has been reinvigorated, following the observation that curative approaches in mental health could not meet all the demands.

### 1.1. Specificity of the Field of Prevention and Mental Health Promotion

With regard to mental health as a field of intervention, it is possible, as with physical health, to distinguish five levels of intervention according to the health status of the beneficiary population (see Figure 1). Each level of intervention refers to an approach involving specific methods of action and specialized professionals.

To ensure this distinction between the different levels of intervention and in order to reduce the confusion surrounding prevention, the WHO recommends that the term prevention be reserved for primary prevention [4,5]. It should also be noted that there are three forms of primary prevention: universal, selective and indicated [4,5,6]. Universal primary prevention targets the general population or a subgroup of the population with no risk factors associated with the disorder in question. Selective primary prevention and indicated primary prevention target a subgroup of the population at risk of developing a mental disorder. Selective prevention targets a population at risk due to family or environmental factors (e.g., an intervention to prevent depressive disorders in children of parents with a mental disorder). The indicated prevention targets individuals at individual risk and/or some warning signs, but with no indication of a proven mental disorder (e.g., intervention for the prevention of depressive disorders in adolescents with malaise).

The principle of primary prevention (prevention of psychological disorders and promotion of mental health) is to act before the problem, by anticipation. It targets populations that are not sick and that are in the living environments of individuals (day-care centers, schools, social centers, companies, retirement homes, homes, natural and urban environments, etc.), and rarely in care settings, as is the case with level three, four and five interventions. Since people targeted by primary prevention are not ill, the approach does not focus on symptomatology or dysfunctional psychological and/or behavioral processes such as identification, psychological counselling or therapy, but rather on the determinants of psychological disorders and positive mental health. Primary prevention therefore aims to modify certain “key” determinants of mental health and/or mental disorders in order to prevent their occurrence and to strengthen positive mental health.

Although, conceptually, mental health promotion is more focused on a positive approach to mental health (to act on the determinants of mental health to increase positive mental health which enables individuals to live a productive life), in practice, the approach and interventions to primary prevention of mental disorders and mental health promotion tend to overlap [4].

Primary prevention and mental health promotion interventions aim to increase protective factors and reduce risk factors associated with mental health (or a specific mental disorder in the case of prevention). They require the mobilization of a broad body of knowledge, particularly on risk and protective factors (which can be modified and are largely associated with mental health) and intervention strategies that are effective in addressing these factors that determine mental health. To be fully effective and efficient, this field of intervention requires an effective articulation between the knowledge produced by scientific research and that derived from experience. This articulation is highlighted in the evidence-based paradigm that is at the heart of the field of prevention and (mental) health promotion. It is based on compelling evidence (evidence-based prevention and mental health promotion).

### 1.2. The Key Role of Psychosocial Competencies in the French National Strategical Orientation Planning of Prevention and Youth Mental Health Promotion: A Historical Perspective

Today, the psychosocial competencies (PSCs) are included in the French national strategy of prevention and youth mental health promotion [7]. Since the early 2000s, PSCs have been included in the French national reports and plans (e.g., suicide prevention program 2011–2014; solidarites-sante.gouv.fr, accessed on 9 September 2022). The emphasis on the prevention of mental disorders and the promotion of mental health among young people through the implementation of validated PSC programs is mentioned in the first national mental health plan of the French Ministry of Health (2005–2007, Ministry of Health). These guidelines and action programs adopted the recommendations made in the report submitted to the Ministry of Health in 2003 for the development of psychiatry and the promotion of mental health [8]. Under the heading “better detection and treatment of psychological disorders in children and adolescents, better promotion of their mental health”, it is proposed to implement mental health promotion programs. Mental health promotion programs mainly focus on the development or reinforcement of protective factors in risk situations and are known in France as “social skills development” (Proposal n°III-03, p. 57, [8]). This strategic direction was linked to the first mental health program developed in 2005 by INPES, the former French Institute of Prevention and Health Education, which then became Santé publique France in 2016. This program was titled: “Facilitating the implementation of actions to promote mental health among young people and combat violence”. A multimodal program in nurseries was planned, including the development of PSCs.

In 2008, the INPES, in partnership with the Réseau francophone international pour la promotion de la santé (International Francophone Network for Health Promotion), published a reference guide on good practice “Risky behaviors and health: taking action at schools” [9]. The guide presents a summary of the international scientific and technical literature on interventions to prevent risk behaviors and their implementation; it highlights several mental health prevention and promotion programs based on the development of PSCs, such as the Norwegian “Bullying Prevention Program”, the school violence prevention program “Mieux vivre ensemble dès l’école maternelle” (Better living together starting from Kindergarden), the primary prevention program for addictive behaviors, “Life Skills Training”, and the Quebec positive mental health program “Zippy’s friends”.

In 2008, as part of a communication campaign to combat substance use among youth, and to remobilize parents who have “given up”, a summary of the literature on parenting and parenting programs was produced by the INPES [10]. It highlighted the effectiveness of some parenting programs based on the development of parents’ (and children’s) PSCs, such as the Triple P program and the Strengthening Families Program (SFP). They have positive effects on parental behavior, the quality of parent–child relationships and the prevention of risky behaviors by children and youth (violence, substance abuse, etc.) on a large scale, in the short and long term. In addition to the national campaign “Parenting and prevention of substance use among young people”, the INPES has therefore decided to conduct an action-research project to test and evaluate the PSC family SFP program. Other action-research projects on the PSC programs, such as the Quebec Brindami program, took place at the same time. However, the INPES experimental PSC projects face many challenges, including issues of transferability, acceptability and implementation. In view of the difficulties encountered by this first evaluation research on PSCs, the experiments were outsourced to French associations for prevention and health promotion with a local presence. The INPES, which then became Santé publique France, supervised the evaluation and provided financial support for the experimentation and initial deployments.

In 2010, another alternative approach was adopted within the INPES to respond constructively to the implementation difficulties that had been encountered. Developed in close collaboration with professionals in the field and academics, a project for PSC reference materials (theoretical and practical) based on available scientific and experiential knowledge was born. This project for PSC guidelines is included in the Ministry of Health’s 2018 National Mental Health Plan [11], which will be implemented in the new French National Strategy based on the PSCs development of youth.

This synthesis article aimed to present the research (EBM/EBPH) of the new French strategy for the development of psychosocial competences of children and youth and the main scientific knowledge on which it is based: evidence-based SEL programs, mental health determinants (PSCs), and new PSC definitions and classifications.

## 2. Methods

### 2.1. The EBM/EBPH Paradigm to Build the New French National Strategy

In general, the term “evidence-based intervention” can refer to two different realities. This dual use can be confusing and often hinders the use of evidence in (mental) health promotion and prevention practices.

Most often, the terms “evidence-based intervention” or “evidence-based program” are used narrowly to refer to an intervention whose efficacy has been demonstrated by experimental evaluative research, and is therefore synonymous with a “validated intervention or program”. An evidence-based health intervention is therefore a “program, practice, principle, procedure, product, pill, or policy that has been shown to improve health-related behaviors, health outcomes or environments” [12]. To be considered effective, the intervention must be formalized in such a way that it can be replicated. Its efficacy must be demonstrated by a rigorous evaluation protocol (i) experimental design, the randomized controlled trial being considered the ‘gold standard’; (ii) demonstrated outcomes must be related to a health dimension or a health problem or a key determinant (not just a secondary determinant such as knowledge of risks or attitude); (iii) outcomes must have been replicated in at least one secondary evaluative study [13]).

In a second case, the concept of “evidence-based intervention” refers to a broader concept, the intervention paradigm that emerged in medicine in the 1990s as “evidence-based medicine (EBM)”, to increase the effectiveness of medical interventions by giving greater weight to evidence from clinical research [14]. In this case, an “evidence-based intervention” is one that fits within the EBM paradigm and uses its methodology.

The concept of EBM was formally established in the early 1990s, although its foundations are much older. In 1992, the Evidence-based Medicine Working Group introduced the notion of EBM and presented this approach as a new paradigm. Sackett, a member of this group and considered one of the founders of EBM, defines it as: “the conscientious, explicit and judicious use of the best evidence currently available to make decisions about the care of each individual patient. The practice of evidence-based medicine thus involves integrating individual clinical expertise with the best available clinical evidence from systematic research. By individual clinical expertise, we mean the skill and judgement that individual clinicians acquire through clinical experience and practice” [15].

Therefore, evidence-based practice requires the integration of three types of knowledge: (1) the most relevant scientific knowledge (or evidence); (2) the experiential knowledge of professionals (i.e., the knowledge and skills derived from the experience of professionals); (3) the experiential knowledge of the recipient audience (or knowledge derived from individual experience). Contrary to the restrictive reading that is often given to EBM, the best available scientific knowledge is therefore only one of the three sources of knowledge necessary for practice (see Figure 2).

The EBM paradigm has since been extended to the areas of health and social care. Therefore, currently we refer to evidence-based (mental) health promotion, evidence-based prevention, evidence-based public health (EBPH), and evidence-based practice in psychology (EBPP) [16]. The transposition of EBM, from the (individual) clinical setting to public health (collective), and in particular to prevention and health promotion, raise various questions [17,18,19]. However, as Jané-Llopis et al. [20], and Smith et al. [16], underlined, these approaches are all rooted in and stem from the EBM approach as presented by Geddes et al. in 1996 [15]. In particular, they recall the need to combine scientific and experiential knowledge. The aim is to make sure that the intervention fits with the context and is also based on the practitioners’ expertise of the population with which they work, which can guide the selection and application of the programs [15,16]. Similarly, Kohatsu et al. highlighted the similarity between evidence-based public health and evidence-based medicine which also needs to integrate the perspectives of community members to develop and implement public health actions [18]. Similar to the evidence-based definition of health promotion proposed by the WHO [16], they also point to the fact that the scientific knowledge to be mobilized is particularly numerous and drawn from many disciplines. Indeed, health promotion is informed by many types of evidence-based studies derived from a range of disciplines. These include epidemiological studies about health determinants, health promotion programs effectiveness trials, ethnographic studies about social and cultural influences on health needs, sociological research about the patterns and causes of inequalities, political science and historical studies about the public policy-making process, as well as economic research about the cost-effectiveness of interventions [16].

There are two key complementary ways to develop and implement an evidence-based mental health prevention and promotion interventions in practice: (1) the transfer and use of an existing evidence-based program that has already been demonstrated to be effective, using an “oven-ready” program that has already been scientifically validated, (2) the transfer and use of scientific and experiential knowledge to develop a new intervention based on evidence, expertise and experiences.

As an evidence-based plan for prevention and youth health promotion, the new French National Strategy requires the use and integration of the best available knowledge and practices produced by researchers and professionals.

### 2.2. The new French National Strategy Based on Literature Reviews

Several literature reviews have been conducted since 2010 and used to build the new French National Strategy on the best scientific knowledge.

In 2011, in order to inform public policy, the INPES conducted a systematic review of current knowledge on validated prevention and mental health promotion interventions for children and adolescents [21].

In 2015, the INPES conducted a narrative review on PCSs in order to better understand the PCSs evidence-based programs and their effects on health and youth development [22].

In 2021, in order to provide a mental health advisory for the COVID-19 crisis, a narrative review was led by the High Public Health Council on the determinants of mental health and validated programs for the prevention and promotion of mental health in health crisis situations [3].

In the first half of 2022, two synthesis documents containing theoretical and scientific knowledge on PSCs were produced by Public Health France [4,5]. They aimed to build a common culture by sharing key current academic knowledge and providing benchmarks for action and decision-making. These documents include an updated definition and categorization of PSCs based on international definitions and current scientific knowledge; a review of the effects of successful PSC programs on health and educational success; a list of factors associated with the effectiveness of the PSC’s interventions such as:-The PSC intervention is structured and focused (set of activities organized and manualized).-The implementation of the CPS intervention is of high quality (quality of the training with clear explanation of mechanisms of action).-The PSC activities’ content is based on the scientific knowledge and evidence-based practices.-The PSC activities are intensive and long term.-PSC activities use experiential methods and are based on positive education principals.

## 3. Results

### 3.1. The Key Role of PSCs in Evidence-Based Programs for Prevention and Mental Health Promotion

In the review conducted by the INPES in 2011, 22 forms of effective prevention and mental health promotion (PMHP) interventions were identified, half of which target the general population (universal prevention), and the other half targeting vulnerable groups (targeted prevention). The majority of the interventions were aimed at young people aged five and above. Half of the interventions were multimodal. These 22 forms of effective PMHP interventions are almost all aimed at the development of the PSCs. Only four do not directly address the development of PSCs: perinatal visit programs for vulnerable parents (such as the nurse–family Partnership), perinatal interventions to prevent depression, physical activity interventions to prevent depression and improve self-esteem, and adult mentoring (such as the Big Brothers/Big Sisters of America). Among the 18 types of effective PMHP interventions involving the development of PSCs, one can distinguish unimodal interventions aimed exclusively at the development of the PSCs of children (four types of interventions) or PSCs of parents (three types of interventions), in the general population or with at-risk groups. Some unimodal interventions seek to develop PSCs online for youth at-risk (indicated prevention) with programs based on cognitive and behavioral therapy models (such as the MoodGym program). Some multimodal interventions target the development of PSCs in children and parents and/or teachers, as well as in the wider educational community. Other multimodal interventions aim to develop PSC in addition to other activities and/or themes such as early cognitive and language stimulation activities, prevention of sexual violence, peer interventions, prevention of violence in schools and in the community, and situations of divorce or bereavement.

The Public Health France reports published in 2022 present the effects of PSC programs on health and in particular on mental health through several literature reviews carried out over the past 10 years [23,24].

The 2015 WHO report shows that PSC programs are an effective strategy for preventing violence among young people [25]. Furthermore, the meta-analysis of Durlak et al., in 2011 [26] synthesizes the results of 213 school intervention studies (n = 270,034 students). The PSC programs studied were based on fairly long interventions, consisting of an average of 40 sessions. The results of this meta-analysis have shown that, compared to students in the control groups, those who participated in PSC programs reported significant improvements in self-esteem, in the quality of peer and teacher relationships, school performance, and the reduction of stress symptoms, anxiety and depression, as well as in the reduction of bullying and violence. In addition, this meta-analysis highlighted the importance of the quality of program implementation: the most effective programs are those that are interactive, use role plays, and provide structured activities to guide students towards specific goals.

A second meta-analysis [27] that identified 82 intervention studies (n = 97,406) examined the long-term effects of PSC programs. Results show that students who have benefited from these programs continue to report positive effects in the assessed dimensions (i.e., self-esteem, positive relationships, academic performance, reducing stress, anxiety, depression, violence and bullying) between six months and four years after the end of the intervention.

A third meta-analysis published in 2012 had already completed the data presented by analyzing the effects of 75 programs on prosocial behavior and reduction in substance use, among other things [28]. This meta-analysis also highlighted the beneficial effects of these programs in these areas.

O’Connor and his colleagues reviewed the literature on promoting school mental health [29], updating previous literature reviews, in particular that of Durlak [26]. The authors identified 29 studies, most of which focused on social emotional learning (SEL). A wide variety of measures were used. Twenty-seven out of the twenty-nine identified varied positive outcomes such as improved wellbeing, social and emotional skills, less anxiety, better anger control, less stress, better concentration, better ability to relax, better problem-solving skills, fewer suicidal thoughts, better knowledge of mental health, and better acceptance of people with mental health issues.

Shortly thereafter, in 2019, a systematic review was published to identify the effects of these programs on adolescents, also taking into account research on older adolescents in previous syntheses. It therefore focuses specifically on middle- and high-school students and aims to identify the effects of interventions on targeted socio-emotional skills [30]. The PSCs identified in this meta-analysis are based, as in previous meta-analyses, on the CASEL model. This synthesis includes 40 studies of 32 programs whose main objective is self-awareness, self-management and interpersonal skills. Half of the programs also focus on awareness of others, while less than half focus on responsible decision making. Most of the research conducted did not assess the effects of the program on the targeted PSCs: less than half of the studies targeting self-regulation skills measured the effects on these skills, and even fewer studies included measures of all the other targeted psychosocial skills. However, when these skills have been measured, the results of these studies show that the programs do indeed develop the targeted skills, as well as psychosocial health. The most significant effects are observed concerning self-awareness and awareness of others and the least significant effects are observed with respect to interpersonal skills. These skills potentially require more training to make lasting behavioral changes, while workshops already allow for greater awareness of self and others. These skills of awareness of self and others have been identified as a useful prerequisite for further development of self-regulation and interpersonal skills [31,32,33]. Indeed, in the synthesis by van de Sande et al., 2019, the results indicate greater significant effects for self-regulation and interpersonal skills when evaluation occurs later after the intervention.

### 3.2. The Key Role of PSCs in the Determinants of (Youth) Mental Health

It is now widely accepted that mental health and mental disorders are due to multiple causes [4,6,27,34,35,36]. Similar to physical health, mental health is determined by a set of bio-psycho-social-environmental factors, some of which are conducive to good health (factors of protection) and some of which are unfavorable (factors of risk).

The report of the Institut National de Santé Publique du Québec [34], which adopts the model developed by McDonald and O’Hara [37], reminds us that mental health depends not only on individual and environmental factors, but also on their interaction. This model also stresses the importance of taking action at all ecological levels: the individual, the immediate environment (family and close friends) or the broader environment. Finally, this model is based on a developmental approach, that is; it recognizes that maturation and development result from the interaction between all individual factors and the environment. In this sense, it recognizes that what we experience during critical periods of development, as well as during periods of transition, can have subsequent effects on an individual’s mental health during their lifetime [34].

The generic factors are of particular interest to the field of mental health prevention and promotion. Factors are considered generic when they are associated with several psychological disorders and positive mental health [34]. These generic factors, which may be related to risk or protection, are usually grouped into three main categories (individual factors, relational and family factors and environmental factors), each of which can be further subdivided into different subgroups (see Table 1).

Among these generic mental health factors, several are relevant to PSCs, either because they are explicitly PSCs (as protection factors, or lack of PSCs as risk factors) (in bold in Table 1) or because they are factors directly associated with PSCs (in italics in Table 1); [3,4,6,21,22,27,34,35,36,38,39,40,41,42,43,44,45,46,47,48,49,50].

The majority of individual mental health protective factors are PSCs or are directly associated with PSCs, such as: sense of belonging and safety, autonomy, self-esteem, organizational thinking and skills, healthy behaviors, academic achievement... Many individual risk factors are closely associated with the PSCs: stress, risk-taking behavior, feelings of helplessness, attention and learning problems, fears and feelings of insecurity, feelings of isolation, impulsivity, opposition, pessimism...

The PSCs are also involved in relationship and family factors. They play a crucial role in parenting skills and, therefore, in the protection and risk factors associated with parents, the parent–child relationship and the relational situation within the family. All risk and protective factors related to peer relations are also associated with the PSCs.

Given the link between the PSCs and the quality of relationships, they are also associated with the social environment, including protective factors such as social support and community cohesion and factors of risk such as rejection and discrimination.

### 3.3. Psychosocial Competencies: A New Definition and Classification for the French National Strategy

Over the past thirty years, the definitions and classifications of PSCs have evolved. Based on the initial definitions proposed by the WHO, integrating classifications produced by international organizations and based on data from the literature and expert opinions, Public Health France documents published in 2022 present an updated definition and classification of PSCs, which aims to propose a common culture on PSCs, and to base interventions on common grounds [23,24].

The first definition of psychosocial competencies was proposed by the WHO Department of Mental Health in the 1990s, particularly in its background paper on life skills programs in schools [51]. The WHO presents these competencies as core competencies with intercultural value. Psychosocial competencies defined as: “a person’s ability to cope effectively with the demands and challenges of daily life. It is a person’s ability to maintain a state of psychological well-being and to demonstrate it through appropriate and positive behavior in their interactions with others, their culture and their environment” [51]. The development of this global psychosocial capacity requires “the strengthening of the individual’s coping resources and personal and social skills” [51]. In schools, this psychosocial development can be achieved by “teaching basic life skills in a favorable school environment” [51]. WHO has defined life skills as “a group of psychosocial and interpersonal skills that help people make informed decisions, solve problems, think critically and creatively, communicate effectively, build healthy relationships, empathize with others, cope with difficulties and manage their lives in a healthy and productive way” [52]. To achieve this goal, the initial WHO classification in the 1990s provides a list of 10 interrelated skills grouped into 5 pairs. In subsequent documents, the WHO refined the scope and definition of PSCs using psychological models available in the early 2000s. This new classification proposes to group PSCs into three broad categories: one group of interpersonal PSCs (social skills), and two groups of intrapersonal or intrapsychic skills (cognitive skills and emotional and self-regulated skills) [52,53]. Each PSC group (social, cognitive and emotional) contains a non-exhaustive list of psychosocial skills and sub-skills.

During the same period, the American organization CASEL (Collaborative for Academic, Social and Emotional Learning), created in 1994 to support PSC learning from kindergarten to high school in the United States, proposed a generic theoretical framework. The framework focuses on key social emotional learning skills (SEL), defined as “the skills, attitudes and values essential to the social and emotional development of young people” [54]. SEL is considered “an integral part of education and human development” and is defined as “the process by which all young people and adults acquire and apply the knowledge, skills and attitudes to develop healthy identities, manage emotions and achieve personal and collective goals, feel and show empathy for others, build and maintain supportive relationships, and make responsible and caring decisions”. Similar to that of the WHO, the CASEL classification is mainly based on Goleman’s theoretical model of emotional intelligence and explanatory models of behavior from cognitive social psychology (such as Ajzen and Fishbein’s theory of planned behavior) [54]. Similar to Goleman’s Emotional Intelligence classification, CASEL proposes a classification based on four broad PSC categories [54] and then five major PSC categories [55]: self-awareness (emotions, thoughts, behaviors, etc.), self-regulation (of emotions, thoughts, behaviors, etc.), social awareness (of others), relationship skills, and the ability to make responsible choices. Each of these 5 basic PSCs includes about 25 sub-PSCs.

More recently, the Organization for Economic Co-operation and Development (OECD), which highlights that the PSCs are major determinants of educational and occupational success, defined social and emotional competencies as “individual abilities that can (a) manifest as regular patterns of thoughts, emotions and behaviors, (b) be developed through formal and informal learning experiences, (c) are important factors in a person’s lifelong socio-economic outcomes” [56]. The identified social and emotional competencies are organized according to the theoretical framework “Big Five” personality trait, which highlights five major personality traits (OCEAN): Openness to experience, Conscientiousness, Extraversion, Agreeableness, and Neuroticism [57]. This theoretical framework of personality, long used by economists, has been used to classify and measure the PSCs in this area of intervention [58]. Five major domains (corresponding to the five personality traits) comprising fifteen PSCs have been proposed by the OECD and used in their latest international survey [56,59].

Maintaining the theoretical framework proposed by the WHO in the 2000s (in particular the classification into three groups of social, cognitive and emotional PSCs) [52], based on all the PSCs mentioned in the main international classifications and based on current scientific knowledge, particularly in psychology, Public Health France (Santé publique France) proposed an updated definition and classification of PSCs (see Table 2); [23,56,60,61,62,63,64,65,66].

Psychosocial competencies are defined as “a coherent and interrelated set of psychological capacities (cognitive, emotional and social), involving knowledge, intrapsychic processes and specific behaviors, which enable improved empowerment, maintenance of a state of psychological well-being, promotion of optimal individual functioning and develop constructive interactions” [23,24].

The updated Santé publique France classification focuses on the so-called key PSCs mentioned in the literature and presented in the successful PSC programs. Thus, nine general PSCs have been identified, each containing two to four specific PSCs (of 21 specific PSCs in total). Of these nine general PSCs, three are cognitive PSCs (the ability to be self-aware, the ability to control oneself and the ability to make constructive decisions), three are emotional PSCs (the ability to be aware of emotions and stress, the ability to regulate emotions and the ability to manage stress) and three are social PSCs (the ability to communicate constructively, the ability to build constructive relationships, the ability to solve problems) (See Table 2).

## 4. Discussion

The literature syntheses carried out for more than 10 years by the INPES and then SpF have made it possible to demonstrate the importance of the PSCs as major determinants of mental health. They demonstrated the effects of effective PSC programs and made them known to French actors and decision-makers.

With the objective of increasing the effectiveness of French interventions in prevention and health promotion, several experiments and evaluations of effective PSC programs (designed in Anglo-Saxon countries) have been implemented. Policy makers have guided public policy that encouraged stakeholders to enrich their practices with these programs and evidence. Some of these experiments and evaluations have made it possible to replicate the results obtained in the developing countries, which has generated a strong motivation among actors, researchers and decision-makers.

However, they have also brought out certain difficulties, including implementation problems in French practice environments, and even more so in a deployment situation. This explains today the very low rate of territorial coverage more than 15 years after the first experiments. As Cambon, Ridde and Alla already pointed out in 2010 [17], this issue is not new and has already been the subject of many publications, but it is particularly relevant today in France. Beyond the evaluative questions and the questions surrounding the demonstration of proof of effectiveness in ecological contexts, the problem of the applicability of the intervention remains. This refers to the extent to which the intervention can be implemented identically in another context [17]. The gap between French practices and effective PSC programs has raised many questions among professionals since the first experiments, as there was no common culture of social and emotional learning in schools and of evidence-based education. Furthermore, as Kohatsu (2004) [18] notes, they did not want to be considered as “passive recipients” of public health interventions. In order to help the development of a common PSC culture, it is necessary to present the theoretical models of the effective PSC programs and their mechanisms of action [17,19], in order for professionals to be associated with the construction of the intervention and to mobilize their expertise, which will lead us gradually towards the integrative paradigm of the EBM or the new EBPH [18].

In order to overcome the divisions still present in France between decision-makers, field actors and researchers [17], and to propose the best possible interventions in the prevention and promotion of mental health to the greatest number of individuals, it was also useful to encourage collaboration between practitioners and researchers and to support the co-construction of interventions based on effective programs. In this way, it is possible to adapt programs culturally and to the specific needs of certain populations, as well as to the curricula. Indeed, the active dissemination of knowledge and the interaction model based on the contributions of researchers and practitioners is essential to dissemination.

As such, the field experience and expertise contributed to advocating for a consensus between all stakeholders and decision-makers in terms of considering the large impact on health, education and socialization that the scaling-up of PSCs may have. At the Convention on Mental Health and Psychiatry (September 2021), the construction of a multisectoral strategy for the deployment of PSCs was announced as a priority measure (Measure N°11) [67]. A steering committee, co-chaired by health and education, brought together representatives from the different ministries and involved out-of-school, sport, student, traineeship, agriculture education, justice and social sectors. This national coordination developed the strategy that aimed at covering all children, in and out of schools and in their different living places, starting at early stages and throughout their growth [68]. The national strategy considered, as first issues, the capacity building and the locally coordinated approach needed in achieving universal coverage, as well as the time required for such. Within the next 15 years, dynamics at the “department” (equivalent district) level have to be strengthened to support coordination, local development and social adaptation according to their priorities. PSCs will be included in initial and continuous training for all professionals in positions of education, in order to favor the inclusion of PSCs in educational practices. Evidence-based interventions will be targeted toward children and youths most in need according to the priorities identified at local level, while more attention and academic support will be given to strengthening the quality of local and integrated interventions. Monitoring and evaluation will be built at three levels: adaptation of national surveys to measure the coverage of PSCs interventions and the ecological correlation with health and educational outcomes, academic support to evaluate promising interventions at local level, and a framework to monitor processes and progress at “department” level. Resources from the different sectors will support the strategy together with the sharing of local resources improved by the coordination mechanisms.

The challenge today is to deploy a large-scale strategy to strengthen PSCs, in an inclusive manner and at all levels, that takes into account the importance of the school environment, that reaches children from an early age almost daily and throughout their schooling, and that becomes a catalyst for the deployment of PSCs among all agencies working with young people. To this end, all the concerned Ministries have signed the strategy to engage in the development of PSCs in children and youth from 2022 to 2037 ([68], p. 18).

This “scaling up” will require support for collaboration and strengthening synergies between institutional, public health, educational and social actors, national and regional agencies, local authorities, researchers, health promotion and prevention stakeholders, and professionals who work with children and youth. National implementation will also include engagement and sharing of the PSC’s latest knowledge and practices to support dissemination of PSC evidence-based interventions and implementation of quality criteria, for effective measures that take account of the scientific and experiential knowledge acquired over the past decades.

## 5. Conclusions

This article demonstrates the critical role that PSCs play in the prevention and promotion of mental health and public health in general. While (positive) mental health is defined as a complete state of psychological well-being by the World Health Organization [1,4,5], it should be noted that, as early as the 2000s, the same organization defined psychosocial competence as “the ability of a person to maintain a state of psychological well-being” [53]. From the outset, the PSCs were presented as the individual ability to achieve this (positive) mental health condition. Thus, given the diversity of risk and protective factors associated with mental health, PSCs represent major individual factors for protecting mental health; they also occupy a prominent place among proximity factors (relationship and family factors) and, therefore, are among the factors related to the social environment. This knowledge of the factors associated with mental health is echoed in the evidence of interventional research accumulated over the past 40 years. Today, most successful prevention and mental health promotion programs are focused on the development of the PSCs, either exclusively or in conjunction with other forms of intervention. Many PSC programs have demonstrated their value in prevention and promotion of (mental) health, which has had lasting effects on well-being, anxiety and depressive disorders, behavioral problems and violence, risk behaviors (substance abuse, risky sexual behavior, etc.), academic achievement etc.

Based on this international knowledge and 20 years of national and local experience, a national PSC deployment strategy was launched in 2021 in France. It is led by the Ministry of Health with many other Ministries’ contributions: Education, Justice, Community Life, and Sport. Two first reports summarizing the state of scientific and theoretical knowledge on PSCs were published by Public Health France in the first half of 2022 to support this deployment strategy and build a common evidence-based culture [23,24]. These documents include an updated definition of PSCs based on the WHO definition in the 2000s, integrating international definitions and classifications (CASEL, OECD, etc.) and based on data from the current literature. In total, nine general and twenty-one specific PSCs were identified: three cognitive PSCs relate to self-awareness, self-control and constructive decision-making; three emotional PSCs are concerned with awareness of emotions and stress, and the ability to regulate emotions and manage stress; finally, three social PSCs allow you to communicate constructively, develop constructive relationships and solve difficulties.

These first reports provide a theoretical framework; they will be supplemented later by practical guides and materials. Thus, Public Health France supports this movement for the development and deployment of PSCs in France, in all situations and from an early age. The Agency brings its expertise and promotes synergies between the various knowledge and actors involved, with the ambition of the French government that all children of the next generation should be trained in PSCs.

## Figures and Tables

**Figure 1 ijerph-19-16641-f001:**
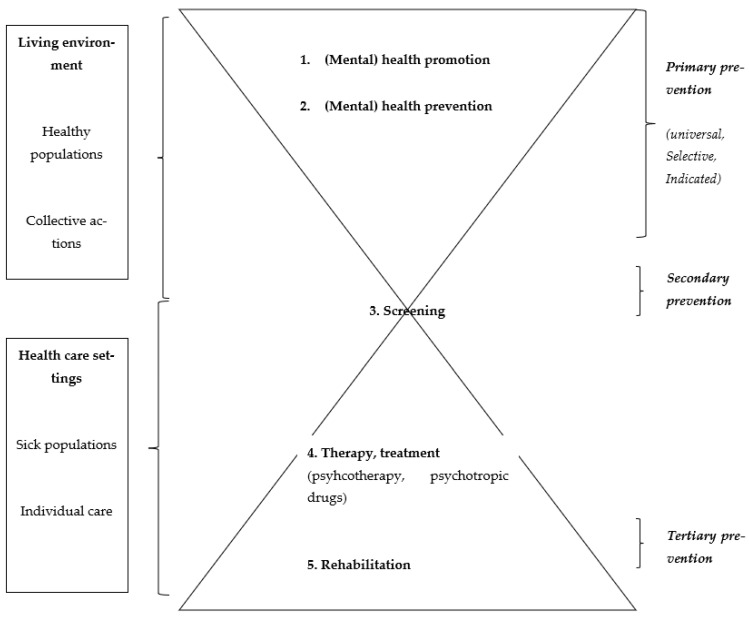
Five levels of (mental) health intervention according to the health status of the beneficiary populations.

**Figure 2 ijerph-19-16641-f002:**
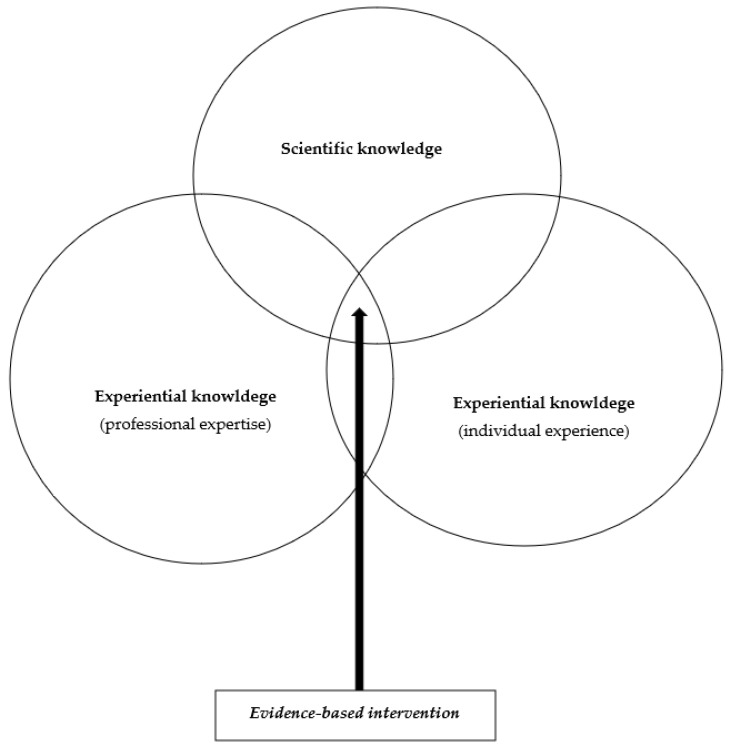
Schematic representation of an intervention according to the evidence-based medicine (EBM) paradigm applicable to evidence-based mental health prevention and promotion.

**Table 1 ijerph-19-16641-t001:** Main generic factors of mental health and mental disorders.

		Risk Factors	Protection Factors
Individual factors	Biological	- Chronic health problems, serious illness - *Unfavorable health behaviors: sedentary lifestyle, substance abuse, unbalanced diet, risk behaviors, sleep problems, excessive use of screens, etc.* - Genetic susceptibility - Problems during pregnancy: *stress, substance use,* maternal malnutrition, infections... - Postnatal problems: low birth weight, complications, head injury...	- Positive health behaviors: physical activity...
	Cognitive	- **Low self-esteem, feelings of incompetence and helplessness** - *Cognitive dissonance* - Problems with cognitive development, language, *attention, learning*…	- **Problem-solving skills** - *Ability to organise* - **Positive self-evaluation: positive self-esteem, sense of self-efficacy...**
	Emotional and social	- *Fears and worries:* related to one’s health, the health of one’s loved ones, one’s own death or that of a loved one - *Sense of vulnerability, of threat* - *Sense of loneliness* - *Sense of frustration* - **Low emotional competencies: impulsiveness...** - **Low social competencies: opposition, aggressive behaviors...** - *Personality difficulties: difficult control of temper, pessimism...*	- Sense of security and confidence - **Adaptability: coping, resilience** - **Emotional competencies: good stress and emotion management, self-control skills...** - **Social competencies: positive communication, pro-social behavior, good conflict management...** - *Optimism* - *Self-reliance* - *Sense of belonging*
	Life events	- Stressful events: violence, abandonment, adverse childhood experiences, school failure, illness of a close relative, hospitalization of a close relative, death of a close relative, separation, social isolation, quarantine, discrimination–stigmatization, financial problems, loss and problems with unemployment... - Stressful working conditions: increased hours and workload, change of duties, lack of staff, lack of security, lack of knowledge and agreement on procedures, inconsistent organizational support, disorganization, lack of guidelines...	- *Academic success* - Good working conditions: effective institutional and organizational support, effective leadership, training, clear communication, valuing management
Relational and family factors	Parent-child relations	- Exposure to *lack of maternal care:* neglect, rejection, lack of affection... - Exposure to *abuse, mistreatment* - Exposure to *inconsistent discipline, role confusion (parent, teacher...)*	- *Positive parenting*: **parenting skills**, *consistent discipline, affection, sense of parental efficacy...* - *Good interaction: attachment and positive early bonding...* - *Early cognitive stimulation*
	Mental health of parents	- Exposure to parental mental and addictive disorders, *parental stress* - Exposure to parental risk and anti-social behavior (criminality...)	
	Family climate and events	- *Intra-family conflicts and violence* - *Communication problems* - Marital difficulties - Single parent, teenager, with low educational level - Adverse events: unemployment, financial problems, divorce, death, imprisonment, etc.	- *Family support*, especially in the case of adverse events
	Relations with peers and friends	- *Social isolation, social disconnection.* - *Rejection by peers: bullying, intimidation, discrimination* - *Deviant peer influence, social pressure*	- *Social support*, support from colleagues and superiors - *Pro-social peer group* (positive peer influence)
Environmental factors	Social, economic and cultural	- Stressful events: wars, disasters... global pandemic, anxiety-generating communication... - *Rejection, discrimination* - Difficult neighbourhoods (violence, access to drugs, delinquency, poverty, unemployment...)	- *Positive interpersonal relationships: social support, social inclusion, community cohesion...* - Opportunities to have a valued social role - Community activities (sports, leisure, culture...) - Access to and positive organization of care and social services - Economic security - Collective rules discouraging violence
	Physical environment	- Housing problems - Overcrowding - Noise	- Security of housing - Green spaces

Note: PSC as protection factors (or the lack thereof as risk factors) are in bold and PSC related factors are in italics.

**Table 2 ijerph-19-16641-t002:** Updated classification of PSCs (proposed by Santé publique France).

Categories	General Competences	Specific Competences
Cognitive competencies	Self-awareness	Self-knowledge (strengths and weaknesses, goals, values, internal discourse...)
		Critical thinking skills (identification of biases, influences...)
		Positive self-evaluation
		Mindful awareness of inner experiences
	Self-regulation	Impulsivity management
		Goal achievement skills (definition, planning...)
	Constructive decision-making	Ability to make responsible choices
		Ability to solve problems creatively
	Emotion and stress awareness	Understanding emotions and stress
		Identifying one’s emotions and stress
	Emotion regulation	Expressing one’s emotions in a positive way
Emotional competences		
		Managing one’s emotions (including difficult emotions: anger, anxiety, sadness...)
	Stress management	Regulating one’s stress in daily life
		Ability to cope with adversity
	Positive communication	Empathetic listening skills
		Effective communication (valuing, clear expression...)
	Positive relationships	Developing social bonds skills (reaching out, making connections, building friendships, etc.)
Social comptences		
		Prosocial attitudes and behaviors (acceptance, collaboration, cooperation, mutual support...)
	Problem-solving	Ability to ask for help
		Ability to be assertive and to say no
		Ability to resolve conflicts in a constructive way

## Data Availability

Not applicable.

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
