# Peer review of "The Key Role of Psychosocial Competencies in Evidence-Based Youth Mental Health Promotion: Academic Support in Consolidating a National Strategy in France"

_ijerph, 2022, doi:10.3390/ijerph192416641_

Round 1

Reviewer 1 Report

Form:

- lign 158, there is a "." to delete : "Most often, the terms "evidence-based intervention" or "evidence-based program" are used narrowly. to refer..."

- lign 303: "which adopts 302 the model developed by [40], reminds ...":  it's strange not to put the name of the authors after "by"

- lign 337: "3.2. PSC Evidence-based Programmes as a Major Strategy for Psychological Disorder Prevention s and Mental Health Promotion": delete the space before the s

- lign 385/385: "A third meta-analysis published in 2012 had already completed the data presented by analysing the effects of 75 programmes on, among other things, prosocial behaviour and reduction in substance use [57]": the place of the "on" in the sentence, need to be reconsidered in the sentence

- lign 462/463: I d'ont understand why the autors use the ? after the verb. It's the second time.

"INPES, and then Santé publique France, followed (?) the evaluation and provided financial support for the experimentation and initial deployments".

First time: lign 430: "Mental health promotion programmes focus mainly(?) on the development or reinforcement of protective factors in risk situations and are known in France as " social skills development".

Also, a quotation mark is missing.

- lign 477, idem: "The National Public Health Plan (2018-2022), adopted (?) by the Interministerial Committee on Health, defined the development of health education and PSCs from an early age as one of the main priorities of the health policy for children, adolescents and young people".

- lign 524: "While (positive) mental health is defined as a complete state of psychological well-being by the World Health Organization [1], [7], [8], it should be noted that, as early as the 2000s, the same organization defined psychosocial competences as "the ability(?) of a person to maintain a state of psychological well-being"

=> add an S to competence to make them plural

- lign 524: "hability" seems to be the correct term

Content

- lign 538-548, in the conclusion, the paragraph repeats a passage already mentioned in the text. Modify slightly?

- In the section 4.2 ("Current situation and future directions"), we find that future directions are not sufficiently emphasized

Author Response

Reviewer 1.

Comments and Suggestions for Authors

Form:

- lign 158, there is a "." to delete : "Most often, the terms "evidence-based intervention" or "evidence-based program" are used narrowly. to refer..."

Response: Thank you, we corrected this.

- lign 303: "which adopts 302 the model developed by [40], reminds ...":  it's strange not to put the name of the authors after "by"

Response: Thank you, this sentence was modified as the introduction section was modified in order to focus more on the specific topic of this article which is to present how the current scientific knowledge on psychosocial competences supported the development of a national strategy of mental health promotion in youth.

- lign 337: "3.2. PSC Evidence-based Programmes as a Major Strategy for Psychological Disorder Prevention s and Mental Health Promotion": delete the space before the s

Response: This sentence was changed as the introduction was entirely revised.

- lign 385/385: "A third meta-analysis published in 2012 had already completed the data presented by analysing the effects of 75 programmes on, among other things, prosocial behaviour and reduction in substance use [57]": the place of the "on" in the sentence, need to be reconsidered in the sentence

Response: Thank you, we corrected this.

- lign 462/463: I d'ont understand why the autors use the ? after the verb. It's the second time.

Response: Thank you, we corrected this.

"INPES, and then Santé publique France, followed (?) the evaluation and provided financial support for the experimentation and initial deployments".

Response: Thank you, we replaced this by “supervised”.

First time: lign 430: "Mental health promotion programmes focus mainly(?) on the development or reinforcement of protective factors in risk situations and are known in France as " social skills development". Also, a quotation mark is missing.

Response: Thank you, we corrected this.

- lign 477, idem: "The National Public Health Plan (2018-2022), adopted (?) by the Interministerial Committee on Health, defined the development of health education and PSCs from an early age as one of the main priorities of the health policy for children, adolescents and young people".

Response: Thank you, this part was modified, thus this sentence was changed.

- lign 524: "While (positive) mental health is defined as a complete state of psychological well-being by the World Health Organization [1], [7], [8], it should be noted that, as early as the 2000s, the same organization defined psychosocial competences as "the ability(?) of a person to maintain a state of psychological well-being"

=> add an S to competence to make them plural

Response: Thank you, this sentence has been modified.

- lign 524: "hability" seems to be the correct term

Response: Thank you, the term was ability.

Content

- lign 538-548, in the conclusion, the paragraph repeats a passage already mentioned in the text. Modify slightly?

Response: Thank you, we removed this repetition.

- In the section 4.2 ("Current situation and future directions"), we find that future directions are not sufficiently emphasized

Response: Thank you for this suggestion. The discussion and conclusion were modified in order to further develop key aspects.

Reviewer 2 Report

It should be Organization and not Organisation for this particular journal. Is positive mental health where an individual is free from any mental health disorders or someone has a disorder and is able to live a productive life? Should 2.2 be called Mental Health Promotion and Mental Health Prevention?"based practice in psychology (EBPP), 197 ... [18]" I do not think the periods should be there. Also, anything under 10 should be spelled out. Also there needs to be a methodology discussed i.e. inclusion criteria of articles, etc. 

Author Response

Reviewer 2

Comments and Suggestions for Authors

It should be Organization and not Organisation for this particular journal.

Response: We modified this throughout the article.

Is positive mental health where an individual is free from any mental health disorders or someone has a disorder and is able to live a productive life?

Response: We agree with the suggestion of the reviewer and modified this sentence as suggested.

Should 2.2 be called Mental Health Promotion and Mental Health Prevention?"based practice in psychology (EBPP)

Response: The introduction section has been rewritten in order to focus more on the specific topic of this article which is to present how the current scientific knowledge on psychosocial competences supported the development of a national strategy of mental health promotion in youth.

197 ... [18]" I do not think the periods should be there.

Response: Thank you, we corrected this.

Also, anything under 10 should be spelled out. Also there needs to be a methodology discussed i.e. inclusion criteria of articles, etc. 

Response: Thank you, we corrected this.